# Accelerating antimicrobial peptide design: Leveraging deep learning for rapid discovery

**Ahmad M. Al-Omari**[1]*, **Yazan H. Akkam**[2], **Ala'a Zyout**[1], **Shayma'a Younis**[1], **Shefa M. Tawalbeh**[1], **Khaled Al-Sawalmeh**[3], **Amjed Al Fahoum**[1]*, **Jonathan Arnold**[4]

1 Biomedical Systems and Informatics Engineering Department, College of Engineering, Yarmouk University, Irbid, Jordan, 2 Medicinal Chemistry and Pharmacognosy Department, Faculty of Pharmacy, Yarmouk University, Irbid, Jordan, 3 Department of Basic Pathological Sciences, College of Medicine, Yarmouk University, Irbid, Jordan, 4 Genetics Department, University of Georgia, Athens, GA, United States of America

* aomari@yu.edu.jo (AMAO); afahoum@yu.edu.jo (AAF)

**Data Availability Statement:** The data link is available within the paper and shown below: https://doi.org/10.6084/m9.figshare.27015307.v1.

## Abstract

Antimicrobial peptides (AMPs) are excellent at fighting many different infections. This demonstrates how important it is to make new AMPs that are even better at eliminating infections. The fundamental transformation in a variety of scientific disciplines, which led to the emergence of machine learning techniques, has presented significant opportunities for the development of antimicrobial peptides. Machine learning and deep learning are used to predict antimicrobial peptide efficacy in the study. The main purpose is to overcome traditional experimental method constraints. Gram-negative bacterium *Escherichia coli* is the model organism in this study. The investigation assesses 1,360 peptide sequences that exhibit anti- *E. coli* activity. These peptides' minimal inhibitory concentrations have been observed to be correlated with a set of 34 physicochemical characteristics. Two distinct methodologies are implemented. The initial method involves utilizing the pre-computed physicochemical attributes of peptides as the fundamental input data for a machine-learning classification approach. In the second method, these fundamental peptide features are converted into signal images, which are then transmitted to a deep learning neural network. The first and second methods have accuracy of 74% and 92.9%, respectively. The proposed methods were developed to target a single microorganism (gram negative *E.coli*), however, they offered a framework that could potentially be adapted for other types of antimicrobial, antiviral, and anticancer peptides with further validation. Furthermore, they have the potential to result in significant time and cost reductions, as well as the development of innovative AMP-based treatments. This research contributes to the advancement of deep learning-based AMP drug discovery methodologies by generating potent peptides for drug development and application. This discovery has significant implications for the processing of biological data and the computation of pharmacology.

**Funding:** This work was funded by the Deanship of Scientific Research and Graduate Studies at Yarmouk University under Grant Number: 80/2023. The funder had no role in study design, data collection, and analysis, decision to publish, or preparation of the manuscript.

**Competing interests:** The authors have declared that no competing interests exist.

## I. Introduction

Antimicrobial peptides (AMPs) are diverse molecules with potent antimicrobial activity against various pathogens, including bacteria, fungi, viruses, and parasites. Due to the extensive number of terminologies, Table 1 shows the abbreviations and their meanings.

With the increasing prevalence of antibiotic resistance and the urgent need for new strategies to combat infectious diseases, there has been a growing interest in designing novel AMPs with enhanced efficacy and specificity [1]. However, traditional development methods are time-consuming, labor-intensive, and often expensive. In recent years, there has been a growing interest in leveraging bioinformatics and computational approaches, powered by the high-performance capabilities of GPUs, not only to predict and screen potential AMPs but also to solve different bioinformatics problems [2–8]. These methods have the potential to accelerate the discovery process, reduce costs, and improve the overall success rate in identifying effective candidates [9]. Machine learning approaches have emerged as powerful tools for rational drug design, focusing on their ability to analyze large datasets, identify patterns, and make predictions. Machine learning has the potential to significantly impact the field of antimicrobial

**Table 1. Abbreviations.**

| Abbreviations | What it stands for |
| --- | --- |
| AMP | ANTIMICROBIAL PEPTIDES |
| ML | Machine Learning |
| STFT | Short time Fourier transform |
| DL | Deep Learning |
| MIC | Minimum Inhibitory Concentrations |
| APD | Antimicrobial Peptide Database |
| CNN | Convolutional neural networks |
| RNN | Recurrent neural networks |
| DRAMP | Dragon Antimicrobial Peptide Database |
| CAMPR3 | Cationic Antimicrobial Peptides Repository Version 3 |
| MARVIN | Molecular Modeling and Visualization of Interactions in Neuropeptides |
| PCA | Principal component analysis |
| NN | Neural Network |
| KNN | K Nearest Neighbor |
| NLP | Natural language processing |
| Conv | Convolution Layer |
| RELU | rectified linear unit |
| Pool | Pooling Layer |
| FC | Fully connected |
| VGG | Visual Geometry Group |
| ResNET | Residual Networks |
| ROC | Receiver Operating Characteristic |
| FN | False Negatives |
| FP | False Positive |
| TP | True Positive |
| TN | True Negative |
| DLP | Deep learning prediction |
| MCC | Matthews correlation coefficient |
| AUC | Area under the curve |
| ACPs | Antimicrobial Cationic Peptides |

peptide design. Machine learning has the potential to accelerate the design of antimicrobial peptides with improved activity and selectivity [10]. Several types of research have been published utilizing deep learning and machine learning to synthesize antimicrobial peptides [10–14]. However, all such research was either not targeted against specific microorganisms or utilized three or four peptide characteristic variables in the design. The activity of AMP is regulated by physiochemical characteristics such as net charge, stereospecificity, hydrophobicity, amphipathicity, secondary structure, peptide length, sequence, and other characteristics [15–17]. These features are also different according to the targeted microorganism (gram-positive or gram-negative). Rational peptide drug design requires an understanding of peptide function groups and the relationship between these groups and the primary and three-dimensional structures. Therefore, in the designing of AMP such factors should be taken into consideration along with the specification of the target microorganism.

*Escherichia coli* is a gram-negative bacterium that shows increased resistance to treatment [18]. More than 20% of *E.coli* isolates were resistant to both first line (ampicillin and co-trimoxazole) and second line (fluoroquinolones) antibiotics. Furthermore, resistance trends in bloodstream infections caused by resistant *E.coli* and Salmonella spp. have remained steady for the last four years [19]. Resistance to antibiotic treatment translates to more difficulty in treating patients, which can lead to an increased rate of hospitalization, more expensive treatment, and an increased mortality rate. This research aims to build deep-learning applications to design novel AMP peptides using *E.coli* as a model and to overcome the challenges driven by traditional experimental methods. Novel antimicrobials are needed due to the rapid rise of pathogen antibiotic resistance, particularly Escherichia coli. Antimicrobial peptide (AMP) design is time-consuming, laborious, and expensive using traditional experimental methods. These methods can't manage the chemical space of potential AMP candidates; hence, new methods are needed to improve prediction accuracy and minimize AMP discovery time and cost.

This study uses the Short-Time Fourier Transform (STFT) and a residual deep learning network to combine machine learning (ML) and deep learning (DL) with time-frequency analysis. This study's contributions:

- Developing a machine learning model that accurately predicts *E.coli* The study demonstrated antimicrobial activity with 92.9% accuracy, utilizing 34 peptide physicochemical parameters.

- Peptide sequences were converted into signal pictures using a deep learning method to improve feature extraction and AMP categorization.

- Using STFT and deep learning together increases feature extraction, making AMP discovery more efficient and understandable.

- Microbial target knowledge and frameworks could save AMP drug research time and money.

A previously constructed AMPs database [20] was utilized, which is composed of 1360 peptides that exhibited activity against *E. coli*, along with their physicochemical characteristics and their activity minimum inhibitory concentrations (MIC). Moreover, this research investigated the effectiveness of modeling feature weight as the amplitude of a sinusoidal signal and the feature itself as the frequency within deep learning networks. Various architectures and training methodologies were explored to evaluate their impact on performance, efficiency, and interpretability. Through extensive experiments on diverse datasets and benchmark tasks, the benefits of this approach were assessed compared to traditional feature engineering and other state-of-the-art techniques.

## II. Literature review

Predicting antimicrobial peptides presents several challenges due to the complex nature of these molecules and the limitations of available data. Understanding and addressing these challenges are crucial for the development of accurate and reliable prediction models of AMP efficacy. Due to the potential therapeutic agents for combating microbial infections, AMPs have been used recently in drug discovery [20]. To predict and identify the activity of AMPs, however, conventional techniques require a substantial amount of time, money, and manpower. In addition, these techniques may not account for the complete peptide spatial structure and physicochemical properties, which may add additional layers of complexity to the prediction. The Antimicrobial Peptide Database(APD) is described [21] as a useful tool for academics and researchers who are interested in antimicrobial peptides. It provides comprehensive data and information on antimicrobial peptides, encouraging further research and advancing education in this field. ADP is also useful for learning about and understanding these peptides and emphasizing the significance of antimicrobial peptides as potential therapeutic agents. APD provides researchers interested in antimicrobial peptides with voluminous data and resources [22].

The function of antimicrobial peptides in preventing microbial infections was discussed in [23]. The article provides insightful information regarding antimicrobial peptides and their potential application in combating infectious diseases. In addition, Mishra and Wang's research focuses on the computational development of effective antibacterial peptides. Regarding the rational design of antimicrobial peptides with enhanced activity and selectivity, the authors discuss the underlying principles and methodologies [24]. The study emphasizes the possibility of using computational methods to generate novel antimicrobial peptides with medical applications. A previously published research [25] provided an overview of the expanding variety of antimicrobial peptide structures and their mechanisms of action. In addition to discussing how antimicrobial peptides interact with intracellular targets, microbial membranes, and immune cells, as well as the structural diversity of these peptides, that study highlighted the adaptability of antimicrobial peptides as a valuable source of novel therapeutics. Recent developments in the design of antimicrobial peptides and novel methods for treating bacteria resistant to multiple drugs were reviewed [26]. To enhance the activity, selectivity, and stability of antimicrobial peptides, they examined a variety of techniques, including antimicrobial peptide modifications, hybridization, and combination therapies. The potential of antimicrobial peptides as effective alternatives to conventional antibiotics is highlighted in the same study [26].

Deep learning techniques have emerged as potential AMP prediction tools. Using multilayered artificial neural networks, deep learning automatically identifies intricate patterns and representations in vast datasets. Therefore, it is possible to develop [27] algorithms that can deduce intricate relationships between peptide sequences and their biological functions. The application of deep learning to AMP prediction has many advantages, including the acceleration of the discovery process by enabling the rapid screening of vast peptide libraries and the identification of additional sequence components. Deep learning algorithms can combine physicochemical properties and structural features to generate more precise predictions. Deep learning models' ability to generalize from observed patterns facilitates the recognition of novel AMPs [28]. RNNs [29], CNNs [30], and attention-based [31] models are a few examples of deep learning architectures that have proven to be effective for AMPs prediction. These structures aid in the development of precise prediction models for AMPs by making use of their special capacities to record sequence patterns, spatial data, and positional importance. Convolutional neural networks (CNNs), on the other hand, are very good at removing spatial

information from input data. CNN architectures, which are widely used in image analysis applications, have been incorporated into peptide sequence analysis [30], where they use filters and pooling techniques to identify regional patterns and hierarchical representations. By applying these filters to peptide sequences, CNNs can identify important local motifs and higher-level structural components that are crucial for the prediction of AMPs. The prediction of antimicrobial peptides (AMPs) has led to increased interest in attention-based models. These models aim to assess the relative significance of various elements within a peptide sequence. These models can effectively identify and give preference to important motifs and structural components that influence the peptide's activity. Additionally, attention-based models can be utilized as generative models, allowing for the creation of novel peptides by assigning varying weights to different positions within the sequence [32]. The advantages of these deep learning architectures arise from their ability to extract spatial and sequential information from peptide sequences. Convolutional neural networks (CNNs) have demonstrated remarkable proficiency in extracting local patterns and higher-level structural features. On the other hand, recurrent neural networks (RNNs) have superior performance in modeling temporal relationships and capturing long-range interactions [33]. Attention-based models provide a more precise understanding of the functional components of a sequence by offering a detailed analysis of critical sections [32]. Several advanced methods for antimicrobial peptide (AMP) prediction use deep learning and other computer models to improve accuracy. One of these new methods is AIPs-SnTCN, which uses stacked temporal convolutional networks to improve antimicrobial peptide identification [34]. With a transformer-based design, DeepAVP-TPPred improves sequence-based antiviral peptide predictions [35]. To better predict antifungal peptides, AFPs-Mv-BiTCN uses multiview learning and bidirectional temporal convolutional networks [36]. In the meantime, the Deepstacked-AVPs model uses deep stacked learning architectures to help classify antiviral peptides more accurately [37]. To predict antimicrobial peptides more reliably, pAtbP-EnC adopts an ensemble classifier technique [38]. Despite these advances, approaches that balance accuracy, interpretability, and computing efficiency are needed [39]. Based on these advances, our research mixes machine learning, deep learning, and time-frequency analysis to optimize AMP prediction accuracy and interpretability.

## III. Material and methods

### III-1. AMP database

There are several possible AMPs databases; the Antimicrobial Peptide Database (APD), The CAMPR database, the Dragon Antimicrobial Peptide Database (DRAMP), and the Database of Antimicrobial Activity and Structure of Peptides (DBAASP). APD is a curated database with extensive information on antimicrobial peptides. It provides AMP sequences, structures, activities, and other relevant information [40]. The CAMPR database provides a selection of experimentally validated AMPs, and CAMPR3 is an updated version of this database. It includes information regarding peptide sequences, activities, sources, and additional relevant annotations [21]. DRAMP is an extensive repository of antimicrobial peptides (AMPs) derived from a variety of sources. It includes data on peptide sequences, structures, functions, and other annotations [41]. DBAASP is an open-access, comprehensive database containing information on amino acid sequences, chemical modifications, 3D structures, bioactivities, and toxicities of peptides that possess antimicrobial properties [42]. Moreover, DBAASP has the largest number of AMPs, around 15 700 peptides. The process of constructing the dataset has already been explained previously [20]. Briefly, the database DBAASP"(https://dbaasp.org) (contains 21743 AMP) was used to extract the sequences and the minimum inhibitory

**Table 2. A single peptide example along with its calculated parameters.** (ENREVPPGFTALIKTLRKCKII) [20].

| Peptides Characteristic | Value | Peptides Characteristic | Value |
|---|---|---|---|
| Atom count | 373 | Partition coefficient (log P) | -8.40 |
| Asymmetric atom count | 26 | LogD | -18.86 |
| Rotatable bond count | 85 | HLB | 150.78 |
| Ring count | 3 | Intrinsic solubility | 17.13 |
| Aromatic ring count | 1 | Refractivity | 664.26 |
| Hetero ring count | 2 | Length | 22 |
| The van der waals volume | 2372 | Normalized hydrophobicity | 0.17 |
| Minimal projection surface area | 294.06 | Net charge | 3 |
| Maximal projection surface area | 494.55 | Isoelectric point | 10.43 |
| Minimal projection radius | 12.18 | Penetration depth | 18 |
| Maximal projection radius | 20.01 | Title angle | 121 |
| Van der waals surface area | 3868.6 | Disordered conformation propensity | 0.05 |
| Solvent accessible surface area | 3235.7 | Linear moment | 0.21 |
| Polar surface area | 1016.85 | Propensity to in vitro Aggregation | 67.42 |
| H-bond donor count | 253.04 | Angle Subtended by the Hydrophobic Residues | 80 |
| Polarizability | 36 | Amphiphilicity Index | 0.84 |
| H-bond acceptor count | 40 | Propensity to PPII coil | 1.04 |

concentration of AMPs against the gram-negative bacteria (*E.coli*) (1360 peptide). Then the developed software big-data bot [20] was utilized to calculate 34 physicochemical characteristics of each sequence using the software package MARVIN resulting in 46240 hits. The 34 physicochemical characteristics that were used are shown in Table 2.

## III-2. Prediction

Two different methods are used to predict peptide sequence activity in Section III-2. The first method feeds preprocessed peptide physicochemical data into a neural network. Training a neural network model to identify peptide sequence activity against *E.coli* is the task. Clean and standardized data on physicochemical parameters, including atom count, solvent accessibility, and hydrophobicity, is used. The direct association between these traits and the peptide's antibacterial activity. Combining data elements, the neural network learns intricate patterns and correlations among antimicrobial features. The second method is more novel since it converts input data into signal pictures for the neural network. This process is like turning numerical data into a visual format for image recognition analysis. The neural network can convert visual input into signal images using convolutional neural network (CNN) architectures, which are known for their pattern detection. By viewing transformed data as images, the model can explore intricate, multidimensional correlations between attributes that may not be apparent when looking at numerical data. Thus, this method improves our understanding of how physicochemical factors affect peptide microbe death.

These two approaches benefit each other. The initial approach employs conventional data processing techniques to construct a fundamental understanding by utilizing unprocessed raw data. The second method's challenging pattern identification is enhanced by signal image conversion and deep learning. The two-pronged approach increases the probability that the model will identify peptides, even if the connections between their physicochemical properties and antimicrobial activity are complex and non-linear. This enhances the precision of predictions. The paper demonstrates the use of advanced machine learning and classic data analysis methods to enhance the accuracy and reliability of peptide activity forecasting. This is achieved by

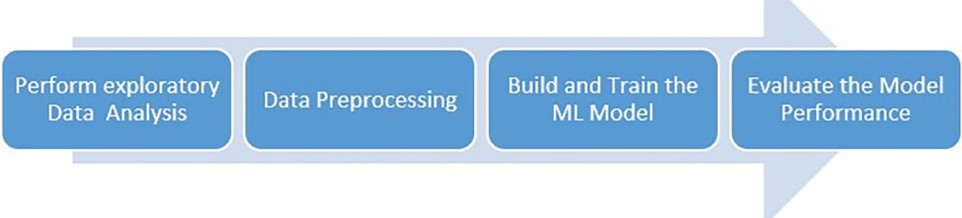

**Fig 1. Overall process of the prediction method.**

utilizing both methodologies. Using typical data processing methods, the initial strategy constructs foundational knowledge from raw data. The second method utilizes deep learning and signal image conversion to identify intricate patterns. This two-pronged approach makes it more likely that the model will find complex, nonlinear links between peptides' ability to kill bacteria and their physical and chemical properties. Deep learning and signal image conversion facilitate the identification of challenging patterns. Modern machine learning and classical data analysis are employed to enhance peptide activity estimates. Both solutions were able to achieve this goal. Fig 1 shows the prediction method's data collection, preprocessing, model selection and training, validation, prediction production, and accuracy and reliability evaluation.

## III-3. Explanatory data analysis

Exploratory Data Analysis (EDA) helps the manuscript analyze data trends, find connections, and prepare data for modeling. EDA was used to investigate the 34 physical and chemical properties of 1,360 peptide sequences to determine from where they were obtained, how they related to each other, and what antimicrobial activity they might have. To simplify future modeling, peptide minimum inhibitory concentration (MIC) features must be identified. EDA checks for outliers, manages missing or chaotic data, and ensures data consistency. Outliers are handled, features are normalized, and missing values are added using the attribute mean. EDA cleans and preprocesses data machine learning models improves usefulness, prepares it for advanced modeling. EDA helps choose and optimize machine learning models, including AdaBoost, KNN, neural network, random forest, CNN, and STFT deep learning for clear visualizations, feature importance ratings, and decision-supporting insights to promote interpretability. The manuscript links unprocessed data to scientific findings, such as new antimicrobial peptides made using EDA. EDA evaluates data structure and distribution to find methods with high specificity or sensitivity. EDA interpretability is stakeholders' ability to quickly understand and share results. Visual techniques, including confusion matrices, ROC curves, and performance matrices, are used to communicate machine learning model results throughout the article. These visuals help experts and non-technical audiences understand peptide classification and features. EDA also determines which physicochemical traits predict antibacterial activity. Important predictors are atom count, solvent accessibility, and polar surface area. Transparency in feature selection and modeling builds stakeholder trust by explaining why specific features are prioritized. EDA outputs show data patterns and correlations to aid decision-making. Identifying *E. coli*-effective peptides guides future studies.

## III-4. Data preprocessing

By constructing a novel database of 34 parameters for 1360 antimicrobial peptides (AMP) sequences from several MARVIN software panels, a novel AMP dataset using big data bot

software was generated [20]. The data collected by the data bot required multiple levels of pre-processing before being fed to the machine learning algorithm, as data quality issues may influence the performance of the model. These problems may include noisy or absent data, duplicate data, irrelevant input features, or outliers. In the initial prediction model, disparate data sources were combined and integrated to form a unified dataset [20]. With principal component analysis (PCA) and a correlation matrix, duplicate rows and superfluous columns were removed to reduce the dimensions of the data set. Next, the data was cleansed by replacing missing values with the attribute mean for all samples of the same class to smooth out noisy data, identifying or removing outliers using z-score = 1.96, and resolving data inconsistencies. Then, the required data transformations, such as mapping nominal data to numerical data and scaling values using a function that maps the entire set of values of a given attribute to a new set of replacement values, were performed so that each old value could be identified by one of the new values. With a threshold of 64, the target value (MIC) was divided into binary categories: MIC values less than 64 indicate active peptides and are converted to 1, whereas MIC values greater than 64 indicate inactive peptides and are converted to 0. In addition, as the final step in data preparation, the ratio of active peptide examples (MIC with label 2) to inactive peptide examples (MIC with label 0) was equalized to address the issue of imbalanced data. As there are more active peptides than inactive peptides, only active peptides with MIC values closer to 0 are retained, as a lower MIC value indicates greater activity. Thus, the distinction between active and inactive peptides' physicochemical properties is readily apparent. A threshold of 64 μg/ml is strategically chosen based on the observed MIC values of various AMPs against clinically relevant pathogens [18]. The antimicrobial peptides must be effective and exhibit low toxicity to host cells, and the MIC threshold of 64 μg/ml has been used as an initial point for such profiling [18]. Fig 2 depicts the disparity between the data in columns A and B.

## III-5. machine learning prediction model

Various machine learning algorithms were employed to predict active and inactive peptide sequences such as AdaBoost [43], K Nearest Neighbor, Neural Network, and Random Forest

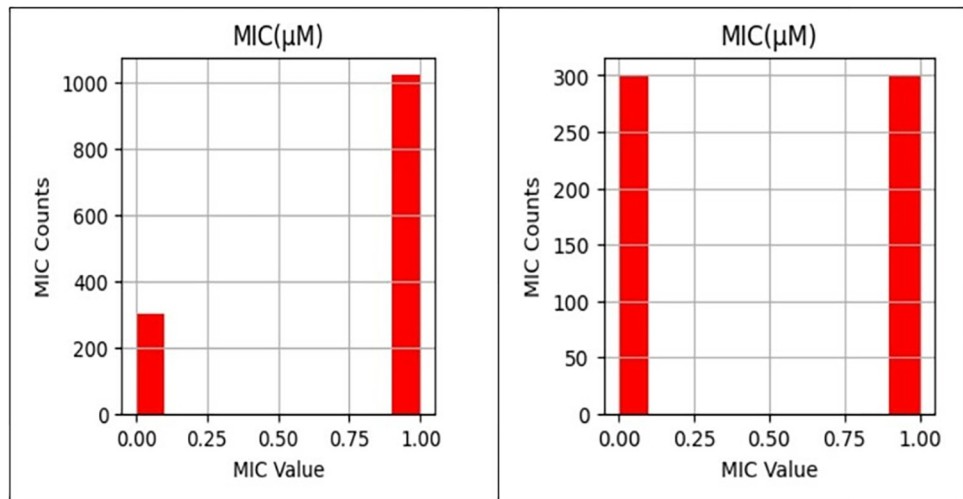

**Fig 2. Left,** the unbalanced MIC target values before performing the data preprocessing, **Right,** the balanced MIC target values after performing the required data preprocessing where the active peptides with MIC value 1.0 were reduced to be equal to the MIC with 0 label counts. The dataset was divided into two subsets randomly: training and validation, with ratios of 80% and 20%, respectively.

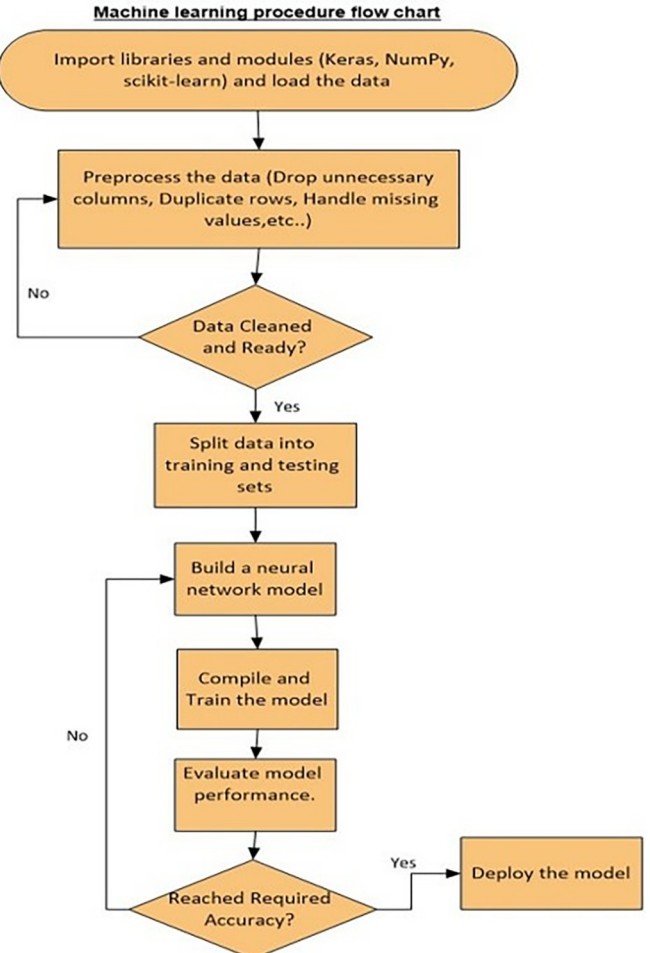

**Fig 3. Machine learning prediction algorithm flow chart.**

using Python with sklearn. Fig 3 summarizes the procedure that was followed to build the machine learning prediction algorithm of the Neural Network as an example.

## III-6. STFT deep learning prediction model

*STFT Deep Learning for Peptide Classification*: The methodology described in this article is geared toward developing and applying a deep learning classification process built on feature modeling and short-time Fourier Transform (STFT) capability. The initial step involves preparing peptide sequences into two types (Effective and Non-effective) for system implementation depending on the MIC value. The feature modeling and STFT are then used to reconstruct the peptide sequence types in 2-D space. The depicted STFTs are then used to create a composite image. As seen in Fig 4, the created images are then transmitted to a deep-learning network to generate classification results. Out of the initial 1,360 peptide sequences, only 1,329 were suitable for use at this stage of the analysis. Among these, 1,046 peptides were classified as effective, while 283 were classified as non-effective. To balance the dataset for further analysis, the number of peptides representing each class was adjusted to match the number in the smaller class. Therefore, after multiple rounds of random shuffling, 283 peptide sequences were selected from each category.

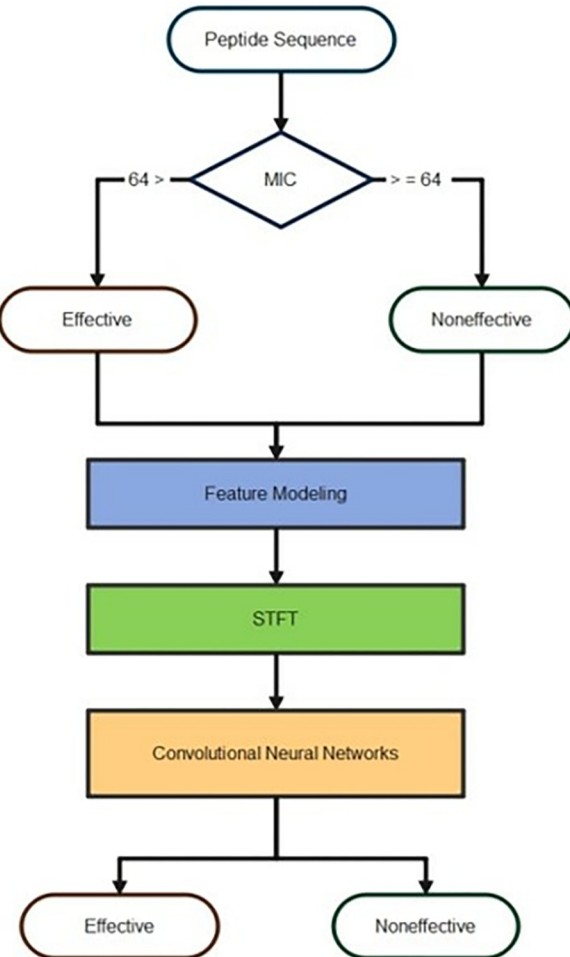

**Fig 4. A block diagram of the proposed approach.**

### III.6.1 Feature modelling

Deep learning networks have revolutionized various fields, including computer vision, natural language processing, and speech recognition. These networks excel at learning complex representations from raw data, allowing them to capture intricate patterns and make accurate predictions. However, their success relies heavily on appropriate feature representations that encode the underlying data characteristics [7, 44].

Traditionally, feature selection and engineering have been crucial steps in improving the performance of machine learning models. Researchers and practitioners invest significant effort in manually designing features that capture relevant information for a given task. While this approach can yield promising results, it is often time-consuming and labor-intensive and may need to generalize better across different datasets or domains [44–46]. Researchers have explored various techniques to automatically learn feature representations directly from raw data to address these challenges and enhance the efficiency of deep learning networks [44–46]. One such promising approach involves modeling the feature weight as the amplitude of a sinusoidal signal and the feature itself as the frequency. This novel approach capitalizes on deep learning networks' inherent frequency-domain analysis capabilities. By assigning the feature weight as the amplitude of a sinusoidal signal, the network can learn to emphasize or attenuate

specific features based on their importance. The feature weight functions act as a modulation factor, modifying the contribution of various features to the final prediction. Moreover, because the feature is represented as a frequency component, the network can recognize the inherent patterns and variations in the data. The network can be trained to retrieve task-relevant information concealed in repetitive or oscillatory patterns by utilizing the frequency domain. Utilizing sinusoidal modulation and frequency-based representation offers several prospective benefits. Manual feature engineering is unnecessary because the network can autonomously extract helpful information from the data. This technique eliminates the inefficiency and subjectivity of feature selection. Due to the weight and frequency of the included features, the network can capture complex interactions and dependencies between features. This representation enables the network to detect patterns that would otherwise go unnoticed. Furthermore, this strategy could improve the interpretability of deep learning networks. The relative importance of various characteristics can be visualized and understood by mapping the feature weight to the amplitude of a sinusoidal signal. This instrument can enhance the model's predictability by providing insights into the underlying data.

The peptide sequence is modeled as follows:

$$p(t) = \sum_{i=1}^{N} W_i \sin 2\pi F_i t \tag{1}$$

Where p(t) is the modeled peptide sequence sample in the time domain, Wi is the value of the feature for that sample ranges from 0–1, and Fi is the unique frequency representing the feature. N is the number of features used to produce the time domain sequence which equals 24 out of the initial 34 features after feature reduction. The frequencies are selected far from aliasing to ensure that features do not overlap and distort their contribution to the produced sequence. All weights are normalized to give them equal important representation to avoid unbalanced weights. Table 3 lists the features that are used to represent the peptide sequence. Fig 5 demonstrates the feature model of effective and ineffective peptides This figure presents a demonstration of feature modeling representation in the time domain for two peptide sequences: (a) the Effective sequence; and (b) the non-effective sequence. The figure demonstrates that each entity possesses unique characteristics that enhance its feasibility and distinguishability from the other entity. Differences are highlighted by the color variations in the figure. While certain commonalities in amplitude could potentially affect the accuracy of

**Table 3. Features used from peptide sequence in the STFT model.**

| No. of Features | Features | No. of Features | Features |
|---|---|---|---|
| 1 | Atom count | 13 | Solvent accessible surface area |
| 2 | Asymmetric atom count | 14 | Polar surface area |
| 3 | Rotatable bond count | 15 | Polarizability |
| 4 | Ring count | 16 | H-bond donor count |
| 5 | Aromatic ring count | 17 | H-bond acceptor count |
| 6 | Hetero ring count | 18 | Partition coefficient (logp) |
| 7 | The van der waals volume | 19 | Logd |
| 8 | Minimal projection surface area | 20 | Hlb |
| 9 | Maximal projection surface area | 21 | Intrinsic solubility |
| 10 | Minimal projection radius | 22 | Refractivity |
| 11 | Maximal projection radius | 23 | Id |
| 12 | Van der waals surface area | 24 | Length |

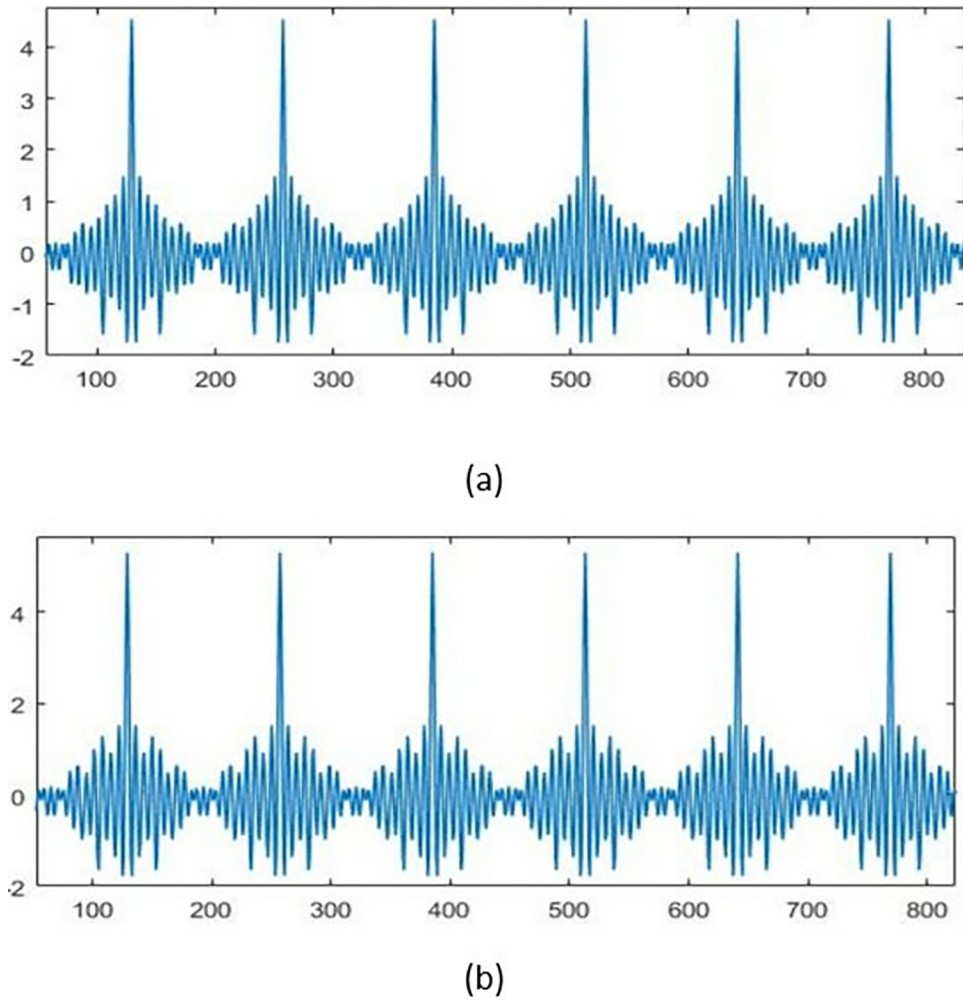

**Fig 5.** Sample of feature modeling representation in the time domain for the peptide sequence: (a) the effective; (b) the non-effective.

future predictions, additional sorts of amplification would be necessary to achieve higher levels of prediction accuracy.

### III.6.2 Short Time Fourier Transforms (STFT)

Sequence analysis is fundamental to numerous disciplines of study and practice [47], such as biomedical signals, image analysis, and signal classification. The Short-Time Fourier Transform (STFT) provides crucial insights into the spectral content of a sequence by decomposing it into its frequency components over time. In this study, we examine how the STFT, when sampled without aliasing, can reveal crucial sequence characteristics. Moreover, we investigate how incorporating a residual deep learning network into the STFT [48] could improve its performance in feature extraction.

To comprehend underlying patterns and extract pertinent properties, the study of sequences frequently involves analyzing their frequency properties. STFT can visualize the spectral content of a sequence in a specific area using time-frequency analysis. The STFT [48, 49] detects time-varying frequencies by dividing the sequence into overlapping brief windows

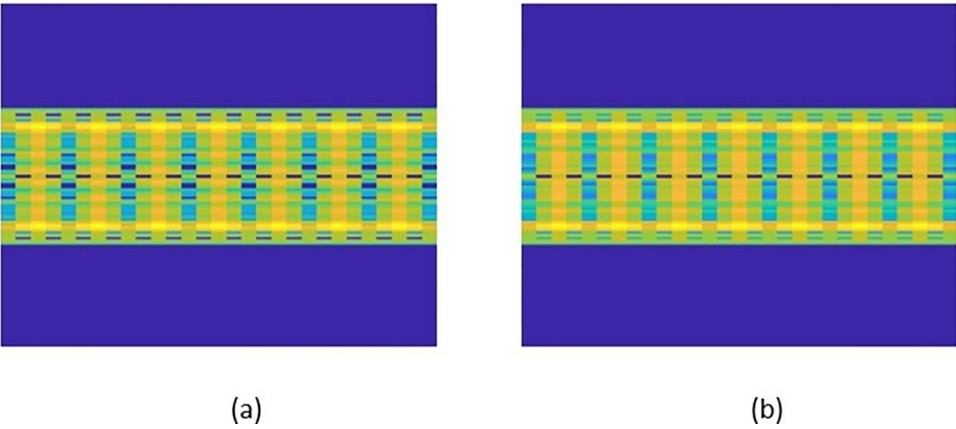

**Fig 6. Shows the sample of the STFT representing two features of the peptide sequence.** (a) the Effective; (b) the Non-effective.

and applying the Fourier Transform to each window. When it comes to elucidating the fundamental characteristics of a sequence, the STFT offers numerous benefits. It provides precise frequency data that can be used to isolate and analyze critical frequencies and monitor their evolution over time. This ability to ascertain resolution is advantageous when evaluating non-stationary data with time-varying spectral characteristics. Second, the STFT permits the detection and localization of frequencies associated with transient events. This localization is crucial for applications such as voice analysis and music processing, where the precise timing and frequency of occurrences are crucial. In conclusion, the STFT enables efficient source separation strategies by mapping individual sources or composite components to distinct time-frequency domain locations. Fig 5 shows the STFT of the sequences shown in Fig 6. Variations between the two images are evident now.

### III.6.3 Convolutional neural network

To further enhance the feature extraction capabilities of the STFT, it is proposed to integrate it with a residual deep learning network. Deep learning networks have demonstrated remarkable performance in various domains by automatically learning hierarchical representations from raw data [3, 5, 6]. By combining the STFT with a residual deep learning network, we can leverage the complementary strengths of both approaches. The STFT provides a time-frequency representation that captures the spectral content, while the deep learning network learns complex feature representations and non-linear relationships within the data [48]. The residual deep learning network architecture, with skip connections and residual blocks, enables the network to capture fine-grained details and residual information in the sequence effectively. This integration allows for extracting more discriminatory and informative features, improving the overall performance of classification through network segmentation, and image denoising. CNN is the best option for image-based classification due to its self-feature learning capabilities and superior classification results on multi-class classification tasks [13, 48]. The components of a CNN include a convolution layer (Conv) with a rectified linear unit (ReLU) activation function, a pooling layer (Pool), and batch normalization. In addition, the last layers include fully connected (Fc), drop-out, SoftMax, and classification output layers. The conv layer contains filters that detect various image patterns (STFT), including edges, contours, textures, and objects. Since ImageNet [30], CNN's architecture has become progressively more advanced. VGG and GoogleNet each have 19 and 22 convolutional layers, whereas ImageNet

only has five layers. Layers cannot be stacked to increase network depth. Due to the "vanishing gradient" problem, training deep neural networks is problematic. Multiple multiplications allow the gradient to be backpropagated to trim levels infinitesimally. As a result, the greater the network depth, the more rapidly it inhibits or clogs. ResNet solves the problem of vanishing gradients using "identity shortcut links." Therefore, ResNet bypasses layers, permitting hundreds of network training layers without degrading performance [7]. ResNet could add a dense layer before the dense layer, be trained from scratch, utilize more substantial data augmentation, and conduct experiments with different learning rates. In this investigation, the Resnet101 CNN model was selected as the optimal model for the proposed method [6].

This paper utilized the ResNet101 model already incorporated in MATLAB® version 2022. ResNet101 is utilized with transfer learning techniques for the final entirely connected layer to be compatible with two classes; the ResNet101 structure is depicted in Fig 4 [48]. The extent of input images for each type is 224*224*3. The data are randomly divided into 70% training and 30% assessment. The model is constructed using the following hyperparameters: adaptive moment estimation (Adam) optimizer, mini patch size of 32, maximal epochs of 60, and initial learning rate of 0.001. Fig 7 shows the architecture for the Res101 Network. To reduce overfitting in the suggested models, a large, carefully selected dataset was used. Since the dataset was large, diversified, and representative of the population, the models were able to generalize to new data. Including a variety of samples and variations prevented models from learning noise or trends from a limited or biased dataset. This protocol improves model resilience and prediction performance across scenarios and real-world applications. The model is robust and capable of detecting any variations that cause it to over fit.

It is apparent from the figures that the Short-Time Fourier Transform (STFT) is a powerful tool for revealing essential features of a sequence, particularly when the sequence is sampled without aliasing. Its ability to capture the time-varying spectral content makes it valuable in various applications. Integrating the STFT with a residual deep learning network further enhances its feature extraction capabilities, enabling the extraction of more informative and discriminative features. This fusion of time-frequency analysis and deep learning holds great promise in advancing the analysis and understanding of sequences in diverse domains. The optimal parameters used for training the machine learning and CNN models are shown in Table 4.

Future research should focus on optimizing the integration of the STFT and deep learning networks to extract the most relevant features and further improve the performance in various sequence analysis tasks.

*Ethics*. Informed consent was not required for this study

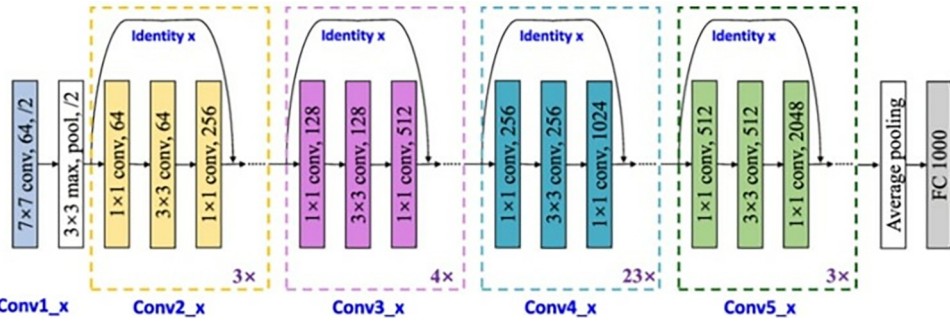

**Fig 7. The architecture of the ResNet101.**

**Table 4. Optimal parameters used for machine learning and CNN models.**

| Algorithms | Optimal parameter |
|---|---|
| Algorithms | **Used Parameters** |
| Random Forest Classifier | n_estimators = 10 to 500, max_depth = 10 to 30, min_samples_split = 10, random_state = 45,class_weight = 'balance' |
| K-Nearest Neighbors (KNN) | neighbors = 5, metric = 'minkowski' p = 4, weights = 'distance', algorithm = 'auto' or 'ball_tree' |
| AdaBoostClassifier | n_estimators = 200 to 500, learning_rate = 0.06 to 0.1, max_depth = 3 to 5 |
| Neural Network | hidden_layer_sizes = (32,64,128,256,1024), activation = 'relu', solver = 'adam', alpha = 0.0001, learning_rate = 'adaptive', Dropout(0.2), loss = 'binary_crossentropy', epochs = 100, batch_size = 10 |
| CNN | Image size = 224*224*3, 70% training and 30% testing, adaptive moment estimation (Adam) optimizer, mini patch size of 32, maximal epochs of 60, and initial learning rate of 0.001. |

## IV. Results & discussion

This section provides an overview of the results obtained from our machine learning prediction model and the Short-Time Fourier Transform (STFT) deep learning production model. In recent years, novel methodologies have been employed to propel the progress of antimicrobial peptide (AMP) design and predict its action. By implementing rigorous analysis and advanced methodology, we demonstrate the effectiveness of our models in correctly predicting AMP activity and using STFT-based strategies to improve pattern identification. The findings emphasize the promise of machine learning in peptide research and demonstrate the revolutionary capabilities of deep learning in unraveling complex biological patterns. Table 5 shows the summary of the machine learning algorithm's evaluation matrix. Due to the high cost of testing active peptides, no inactive peptides must be misclassified as active. Therefore, we focused on reducing false positives and increasing specificity, and the Random Forest algorithm achieved the lowest number of false positives (4) and the highest specificity (0.86).

Regarding the active peptide class, the sensitivity of the AdaBoost algorithm was the highest among all other algorithms and this implies that to use the AdaBoost algorithm for the prediction if the interest was to reduce the false negative rate. Although the performance of the Random Forest and the AdaBoost algorithms were acceptable and helped to distinguish between active and inactive peptides, another novel approach based on STFT deep learning prediction (STFT-DLP) has achieved a higher performance yet.

Various machine learning algorithms were employed as it is shown in Table 5 to achieve the best prediction for the antimicrobial peptides. Figs 8 and 9 display the ROC and confusion matrix for both algorithms corresponding to the highest achieved F1 score.

**Table 5. The performance matrix of different machine learning algorithms.**

| Algorithms | Classes | TP | FP | FN | TN | Precn | F1_Score | Specy | Recall | MCC | Accuracy |
|---|---|---|---|---|---|---|---|---|---|---|---|
| **AdaBoost** | 1 | 46 | 15 | 16 | 44 | 0.75 | 0.75 | 0.75 | 0.74 | 0.49 | 0.74 |
| | 0 | 44 | 16 | 15 | 46 | 0.73 | 0.74 | 0.74 | 0.75 | 0.49 | |
| **Random forest** | 1 | 39 | 8 | 23 | 51 | 0.83 | 0.72 | 0.86 | 0.63 | 0.51 | 0.74 |
| | 0 | 51 | 23 | 8 | 39 | 0.69 | 0.77 | 0.63 | 0.86 | 0.51 | |
| **Neural Network** | 1 | 40 | 22 | 22 | 37 | 0.65 | 0.65 | 0.63 | 0.65 | 0.27 | 0.64 |
| | 0 | 37 | 22 | 22 | 40 | 0.63 | 0.63 | 0.65 | 0.63 | 0.27 | |
| **K-Nearest Neighbor** | 1 | 44 | 17 | 18 | 42 | 0.72 | 0.72 | 0.71 | 0.71 | 0.42 | 0.71 |
| | 0 | 42 | 18 | 17 | 44 | 0.70 | 0.71 | 0.71 | 0.71 | 0.42 | |

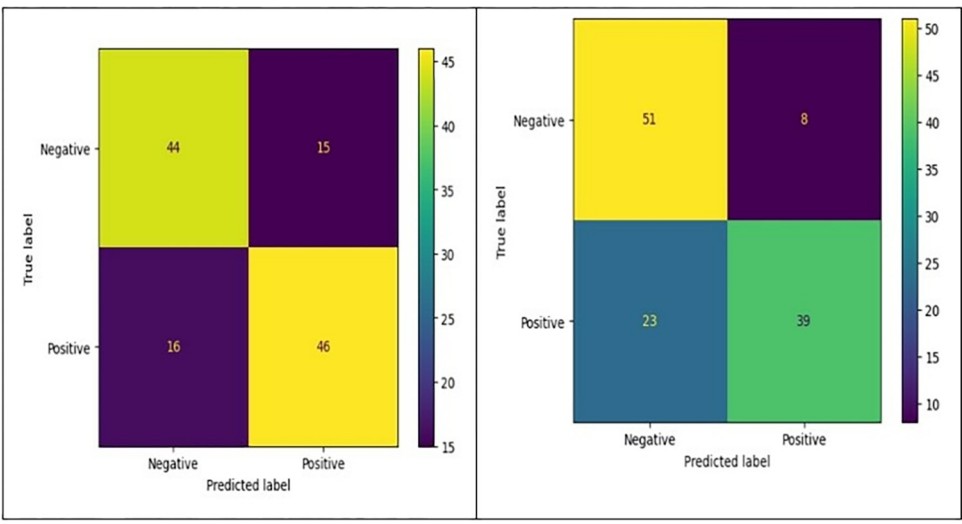

**Fig 8. Left**, AdaBoost classifier confusion matrix of the active peptides (class 1) shows that there are 46 TP and 44 TN were predicted correctly. However, the algorithm predicted 16 FN and 15 FP. **Right,** Random Forest classifier confusion matrix of the active peptides (class 1) shows that there are 39 TP, and 51 TN were predicted correctly. However, the algorithm predicted 23 FN and 8 FP.

For the STFT-DLP method, various performance metrics were utilized to assess the algorithm's effectiveness. Within the existing literature, commonly employed metrics include accuracy, sensitivity, specificity, precision, the F1 score, and the Matthews correlation coefficient (MCC). Accuracy is calculated by dividing the number of correct predictions by the total number of cases in the dataset. A test with high sensitivity can accurately detect the presence of a condition, yielding a substantial number of true positives and minimizing false negatives. This type of test is precious when the medication being assessed is highly effective with minimal side effects. Conversely, the test produces many true negatives and only a few false positives. Precision, also called the "positive predictive value," measures the proportion of relevant examples among the retrieved examples. Recall, which is equivalent to sensitivity, quantifies the percentage of relevant examples that were successfully retrieved. Therefore, both precision and recall are influenced by the concept of relevance. The MCC represents a contingency matrix technique for computing the Pearson product-moment correlation coefficient between actual and predicted values. It serves as an alternative metric that remains unaffected by the issue of imbalanced datasets. Lastly, the F1 score denotes the harmonic mean of accuracy and recall. The confusion matrix, illustrated in Fig 10, provides key metrics regarding the model's performance. Specifically, it showcases a sensitivity of 95.3%, which means that 81 out of 89 images were accurately classified. The precision stands at 91%. Additionally, among 89 images, eight images were incorrectly identified, resulting in a sensitivity of 90.6% and a positive predictive value of 95.1%. Overall, the model achieves an accuracy of 92.9% across both classes. Table 6 summarizes the performance matrices using the proposed approach. It can be observed that the metrics are closely related indicating that the classification approach is robust and precise. Another important metric is the area under the curve (AUC). Fig 11 shows the AUC for both effective and non-effective including the receiver operating curve (ROC). The values are very close to 1 indicating that the classification is accurate and reproducible. The manuscript reports the accuracy of two distinct models for predicting the activity of antimicrobial peptides, with a focus on binary classification using a specific MIC threshold (MIC = 64 μg/ml) to categorize peptides as either active or inactive. The models achieved

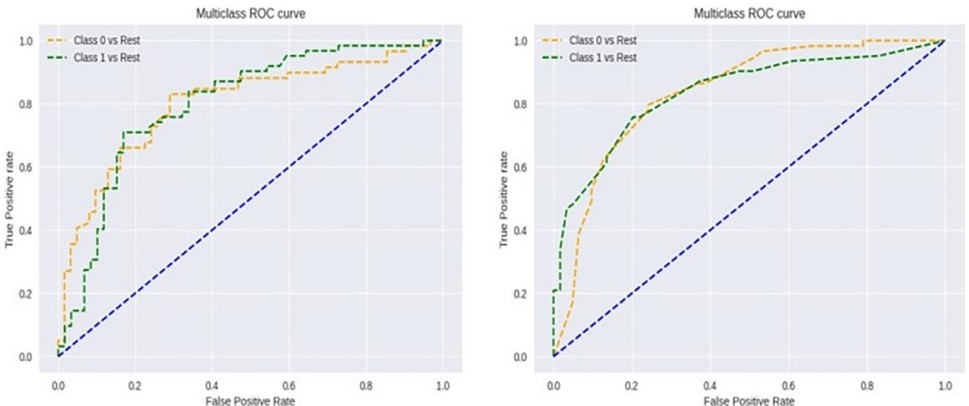

**Fig 9. Left**, AdaBoost algorithm ROC; **Right,** Random Forest algorithm ROC.

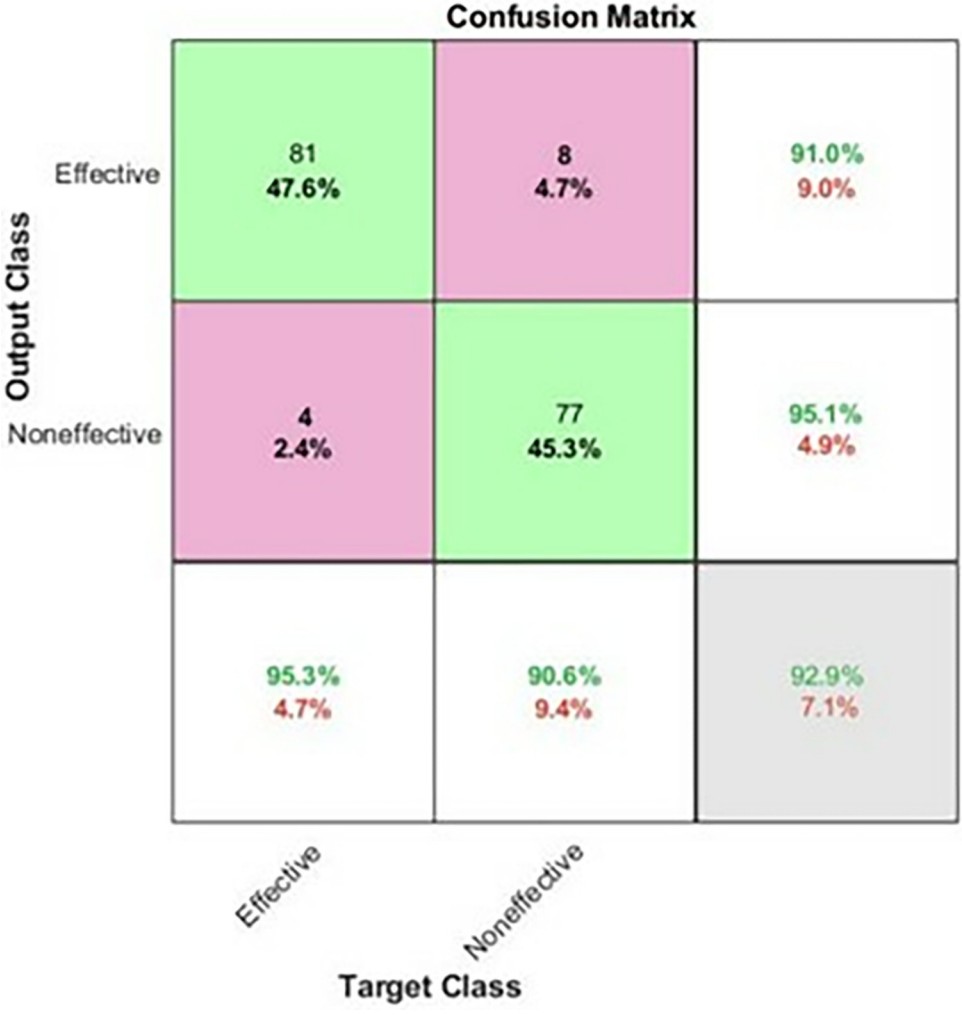

**Fig 10. The confusion matrix for the testing data.** The precision values are represented in the third column, while the sensitivity values are illustrated in the third row of the matrix.

**Table 6. Result analysis with distinct measures from two classes.**

| Classes | Metrics % | | | | | | | | |
|---|---|---|---|---|---|---|---|---|---|
| | TP | FP | FN | TN | $Prec_n$ | $F_{Score}$ | $Spec_y$ | $Recall_l$ | MCC |
| Effective | 81 | 8 | 4 | 77 | 91.0 | 93.1 | 90.6 | 95.3 | 86 |
| Noneffective | 77 | 4 | 8 | 81 | 95.1 | 92.8 | 95.3 | 90.6 | 86 |
| $Accu_y$ | 92.9 | | | | | | | | |

accuracies of 74% and 92.9% respectively, which are metrics sensitive to the chosen threshold. To reduce threshold sensitivity, Tables 5 and 6 and Figs 9 and 11 were included. These metrics consider the trade-off between sensitivity (true positive rate) and specificity (false positive rate), or precision and recall, respectively, making them more reliable indicators of a model's ability to generalize across different scenarios.

Comparative analysis was conducted against a variety of state-of-the-art methods in to verify the efficacy of the proposed deep learning-based model for antimicrobial peptide (AMP) prediction. Table 7 demonstrates the comparative analysis results with state of the art methods.

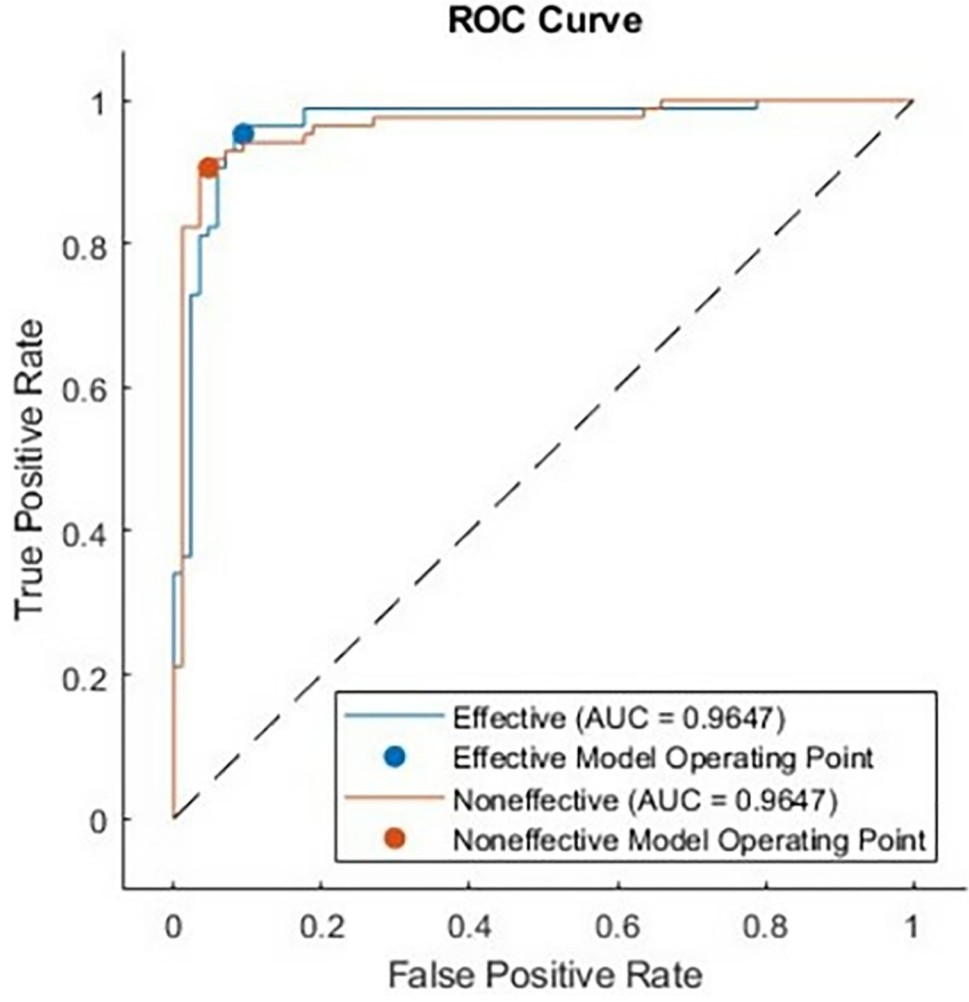

**Fig 11. The ROC curve for testing data.**

Table 7. Comparative performance metrics.

| Method | Accuracy (%) | Precision (%) | Recall (%) | F1-Score (%) | Specificity (%) | AUC-ROC Score |
|---|---|---|---|---|---|---|
| **Proposed Model (Deep Learning + STFT)** | **92.9** | **91.0** | **95.3** | **93.1** | **90.6** | **0.95** |
| **Random Forest (Traditional ML) [28]** | **74.0** | **83.0** | **63.0** | **72.0** | **86.0** | **0.86** |
| **AdaBoost (Traditional ML) [28]** | **74.0** | **75.0** | **74.0** | **75.0** | **75.0** | **0.75** |
| **CNN (Convolutional Neural Networks) [10]** | **85.0** | **87.0** | **80.0** | **83.0** | **82.0** | **0.88** |
| **Attention-Based Model (Hybrid DL) [11]** | **89.0** | **90.0** | **85.0** | **87.5** | **88.0** | **0.91** |
| **Ensemble Learning Model (SAMP) [Feng et al., 2024] [50]** | **88.5** | **89.1** | **87.8** | **88.4** | **87.0** | **0.90** |
| **StackDPPred (Ensemble Learning) [Arif et al., 2024] [51]** | **90.2** | **91.4** | **88.9** | **90.1** | **89.3** | **0.92** |

This comparison encompasses hybrid approaches, sophisticated deep learning architectures, and traditional machine learning models. The proposed deep learning model, which combines a residual deep learning network with Short-Time Fourier Transform (STFT), was tested.

The STFT-based deep learning model outperforms the others on several evaluation criteria. The suggested model is 92.9% accurate, outperforming Random Forest and AdaBoost, which are 74% accurate, Convolutional Neural Networks (CNN), which are 85% accurate, and attention-based models, which are 89% accurate [10, 11, 28]. Additionally, the proposed model outperforms previous ensemble learning methods such as SAMP [50] and StackDPPred [51], which show 88.5% and 90.2% accuracy, respectively. Additionally, the proposed model has 91.0% precision and 95.3% recall. These scores indicate better true positive and false positive detection. However, the SAMP and StackDPPred ensemble learning models have poorer precision and recall. This shows the model's active peptide detection accuracy. With a higher F1-score of 93.1%, the model strikes the right mix between precision and recall. This score exceeds StackDPPred (90.1%) and SAMP (88.4%), indicating strong reliability in applications that use both criteria.

This model has a very high Area Under the Receiver Operating Characteristic (AUC-ROC) score of 0.95, which means it can distinguish items better than CNN (0.88), attention-based models (0.91), SAMP (0.90), and StackDPPred (0.92). This score shows the model's superior ability to differentiate active and inactive peptides. Several factors contributed to the proposed model's superior performance. The Short-Time Fourier Transform (STFT) can turn the basic parts of peptides into signal representations. This lets us find complex patterns that older or more complex models might have missed. Using Short-Time Fourier Transform (STFT) and residual deep learning networks also makes the process of finding relevant features and putting them into groups a lot better. Ultimately, the proposed approach accelerates the process of identifying antimicrobial peptides (AMPs), resulting in time and cost savings compared to computational and traditional methods. The results demonstrate that the deep learning model utilizing Short-Time Fourier Transform (STFT) performs very well and exhibits remarkable promise for peptide categorization. Machine learning, physicochemical properties, sequencing, structural modeling, molecular dynamics simulations, and hybrid approaches were employed to predict AMP activity. Each technique is subject to certain limitations [52].

ML approaches include gradient boosting, neural networks, random forests, and support vector machines (SVMs). Machine learning requires diverse, high-quality training data. The model may not generalize well or provide erroneous predictions if the dataset is biased or lacks peptide group representation [52]. ML algorithms misread biological processes that affect predictions [53]. Physicochemical approaches emphasize secondary structure, amphipathicity, hydrophobicity, and charge for antibacterial action [54]. These methods may not accurately depict peptide activity's complexity and lack of selectivity, resulting in false positives and negatives [55]. This may reduce prediction accuracy because these methods may overlook peptide-

target interaction changes. Antibacterial themes and patterns can be found using AMP amino acid sequence analysis [54]. Sequence-based techniques require datasets that may not represent AMP diversity [56]. These approaches do not anticipate the action of novel peptides; therefore, there are few attractive possibilities [57]. Without sequence-based prediction criteria, findings may be inconsistent [57]. Molecular dynamics simulations and structural modeling show AMPs' conformational dynamics and target interactions. These methods are too computationally and resource-intensive for ordinary applications [54]. Data quality determines structural model validity, and initial structure faults might generate misunderstandings [54]. Many molecular dynamics simulations have too simplistic assumptions and may not fully capture biological system complexity, making them unreliable predictors [54]. Hybrid methods with various prediction algorithms improve AMP activity prediction accuracy and resilience [58]. Integrating several data types may cause compatibility and interpretation issues. Hybrid methods are difficult and require plenty of computational power. Multi-model reliance may hinder decision-making if models disagree on projections [58].

The findings of this work reveal the potential benefits of deep learning techniques in AMP medication discovery. Significantly, the utilization of these methodologies not only results in efficiencies in terms of time and money but also expedites the production of highly effective antimicrobial peptide (AMP) medications. As mentioned above, the results significantly contribute to the field of deep learning, namely in areas of utmost significance, such as pharmaceutical calculations and the interpretation of biomedical data. The significance of precise and understandable feature representations in many circumstances highlights the fundamental worth of this research. Furthermore, this study highlights the integration of the Short-Time Fourier Transform (STFT) with a residual deep learning network to improve the STFT's effectiveness in extracting significant and distinguishing characteristics. The integration of time-frequency analysis and deep learning enhances the capacity to enhance sequence analysis across multiple academic fields. The potential for enhancing analysis and comprehension in various contexts becomes apparent through utilizing the synergistic effects between different approaches.

The convergence of short-time Fourier transform (STFT), and deep learning presents a compelling area for further scientific investigation, highlighting the need for significant focus on optimizing their integration. The primary obstacle in efficiently extracting relevant features to enhance the performance of sequence analysis jobs necessitates a more comprehensive examination. Future research should investigate various architectures and training methods to understand better how these choices affect performance, efficiency, and interpretability. The potential impact of the synergistic interaction between Short-Time Fourier Transform (STFT) and deep learning in sequence analysis is significant. This collaboration can bring about a transformative shift, transcending traditional disciplinary boundaries and leading to novel insights and advancements in knowledge [59, 60]. Various databases and prediction methods were developed to predict and characterize AMPs. Common databases are LAMP [61], CAMPR3 [62], APD [63], and DBAASP [64]. While, AntiTbPred [65], AntiBP3 [66], and LMPred [67] are the common methods to predict the antimicrobial activity, and they differ in their specific focus and underlying algorithms. The CAMPR3 database also has prediction tools. AntiTbPred targets tuberculosis-related peptides, which makes it highly specialized but limited in its ability to target other bacteria. AntiBP3 and LMPred are used in more bacteria, despite not being as pathogen-specific. CAMPR3 may also produce false positives and require experimental validation. Many prediction systems, such as AntiBP3 and LMPred, use sequence-based properties without structural data. The three-dimensional structure of peptides affects AMP activity; therefore, ignoring structural data can lead to erroneous predictions. AMP prediction is limited by experimentally validated sequences [68]. Numerous

prediction models rely on generic antibacterial activity rather than specific mechanisms of action. Assessing peptide efficacy against specific illnesses using this method can be deceptive [69]. When focusing on physicochemical properties, peptide structural conformation and biological dynamics may be disregarded [70]. Antimicrobial peptides (AMPs) have become a viable avenue in the fight against infectious diseases due to their strong effectiveness against a wide range of pathogens. This study provides a compelling illustration of the transformative capabilities inherent in machine learning and deep learning techniques to create new antimicrobial peptides (AMPs) that exhibit enhanced activity and selectivity. This study aims to expand the scope of peptide design by embracing a more comprehensive range of peptide features, explicitly focusing on microorganism-specific physicochemical attributes. This is achieved by systematically resolving the limits encountered in previous research efforts. Using deep learning models for classifying antimicrobial peptides represents a significant departure from conventional experimental approaches, as it effectively bypasses the laborious and resource-intensive nature associated with these traditional methods. Therefore, future research will focus on advanced signal processing methods, including wavelets, bispectral, bicoherence, and their alternatives [71]. Furthermore, combinational machine learning and deep learning methods can improve perception and enhance performance measures [72].

Nosocomial infections and mortality have been on the rise worldwide for the past few decades due to the misuse of antibiotics generating multidrug-resistant bacteria. Therefore, the development of new antibacterial drugs is a crucial request globally [73]. There are several approaches to discovering drugs such as; serendipity [74], chemical modifications of known drugs or natural products [75], expensive screening of natural and synthetic compounds [76], and de novo or rational drug design [77]. Recently, artificial intelligence has invaded drug discovery, where AI was used to predict the 3D structure of proteins, drug–protein interactions, and design molecules (de novo) [77]. However, It has been reported that up to 50% of the approved drugs were from either directly or indirectly natural products [77]. Unfortunately, it's expensive and time-consuming to screen natural products. Host defense molecules known as antimicrobial peptides (AMPs) are ideal candidates due to their ability to permeabilize and disrupt the bacterial membrane, regulating the immune system, broad activities, and limited resistance [78]. Most AMPs are discovered from natural sources or via screening [79]. De novo drug design is also used for designing AMP, but it has some drawbacks such as the structure of the target protein should be known, and limited to small-size molecules. Moreover, Genetic algorithms [80], and linguistic models [81] have also been used to generate antimicrobial peptides. However, such approaches were hindered due to the small number of characterized peptides and due to limitations in the algorithms used at that time [82]. An improvement to this work could be leveraged by using machine learning algorithms to predict the experimental minimum inhibitory concentration of peptide values rather than the binary classification of AMPs. Some recent work has been done in this regard on AMPs prediction [83]. Exposing this work to predict MIC values for AMPs and ACPs is a promising area for future work. This article emphasizes the significance of conducting experimental research in the domains of medication and antimicrobial peptide design. The importance of performing in vitro or in vivo analyses to validate results is highlighted. Theoretical considerations related to identifying potent antimicrobial peptides were also highlighted. Further, it explores computer modeling, algorithm development, and prediction methodologies. It is worth mentioning that the prediction relied on laboratory in vitro data (MIC). Instead of just studying one type of bacteria, like E. coli, future work will include a full research study that uses in vitro assays to test the cytotoxicity and antimicrobial properties of newly designed antimicrobial peptides against many different types of bacteria. Additionally, it aims to explore the potential of using

in vivo models to assess the safety and efficacy of the anticipated antimicrobial peptide candidates.

## V. Conclusion

This study introduces a novel method for expediting the development of antimicrobial peptides (AMPs) using deep learning techniques. The proposed model effectively predicts the activity of AMPs by integrating machine learning and deep learning methodologies. Two main strategies were used: the first used pre-calculated physicochemical properties of peptides in a machine-learning classification method; the second turned these properties into signal representations that were processed by a deep learning neural network. The model achieved a combined accuracy of 74% for the machine learning model and 92.9% for the deep learning model in predicting AMP activity, demonstrating a high level of accuracy. These results underscore the potential of deep learning-based methods to improve the accuracy of predictions and substantially reduce time and costs in the discovery process of AMPs. The results of the study establish a robust foundation for future research in drug discovery, with a particular emphasis on the development of effective AMPs against resistant microorganisms. In order to enhance the efficacy, efficiency, and interpretability of these models, future research would investigate supplementary deep learning architectures and training methodologies. In addition, this method could be shown to be more useful and effective in the larger field of antimicrobial drug design by being applied to different types of pathogens.

## Author Contributions

**Conceptualization:** Ahmad M. Al-Omari, Yazan H. Akkam, Amjed Al Fahoum.

**Data curation:** Ahmad M. Al-Omari, Yazan H. Akkam, Ala'a Zyout, Shayma'a Younis, Amjed Al Fahoum.

**Formal analysis:** Ahmad M. Al-Omari, Yazan H. Akkam, Ala'a Zyout, Shayma'a Younis, Amjed Al Fahoum.

**Investigation:** Ahmad M. Al-Omari, Shefa M. Tawalbeh, Khaled Al-Sawalmeh, Amjed Al Fahoum, Jonathan Arnold.

**Methodology:** Ahmad M. Al-Omari, Yazan H. Akkam, Ala'a Zyout, Amjed Al Fahoum.

**Project administration:** Ahmad M. Al-Omari, Amjed Al Fahoum.

**Resources:** Ahmad M. Al-Omari, Shayma'a Younis, Amjed Al Fahoum.

**Software:** Ahmad M. Al-Omari, Ala'a Zyout, Shayma'a Younis.

**Supervision:** Ahmad M. Al-Omari, Amjed Al Fahoum.

**Validation:** Ahmad M. Al-Omari, Yazan H. Akkam, Shefa M. Tawalbeh, Khaled Al-Sawalmeh, Amjed Al Fahoum, Jonathan Arnold.

**Visualization:** Ahmad M. Al-Omari, Shefa M. Tawalbeh, Khaled Al-Sawalmeh, Amjed Al Fahoum, Jonathan Arnold.

**Writing – original draft:** Ahmad M. Al-Omari, Yazan H. Akkam, Ala'a Zyout, Shefa M. Tawalbeh, Khaled Al-Sawalmeh, Amjed Al Fahoum.

**Writing – review & editing:** Ahmad M. Al-Omari, Yazan H. Akkam, Amjed Al Fahoum, Jonathan Arnold.

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
