## [Decision Letter · Decision Letter 0]

12 Aug 2024

PONE-D-24-16637Accelerating Antimicrobial Peptide Design: Leveraging Deep Learning for Rapid DiscoveryPLOS ONE

Dear Dr. Al-Fahoum,

Thank you for submitting your manuscript to PLOS ONE. After careful consideration, we feel that it has merit but does not fully meet PLOS ONE’s publication criteria as it currently stands. Therefore, we invite you to submit a revised version of the manuscript that addresses the points raised during the review process.

The manuscript addresses a significant and timely topic, yet several aspects need improvement. The quality and presentation of figures, the clarity and flow of the abstract, and the overall consistency in data reporting require attention. Reviewers also emphasize the need for a stronger discussion on the model's evaluation metrics, handling of overfitting, and comparison with state-of-the-art methods like AtniTbPred etc. Additionally, enhancing the accessibility of the study through a GitHub repository or web server could significantly benefit the research community. Addressing these comments will substantially improve the manuscript's quality and impact.

We look forward to receiving your revised manuscript.

Kind regards,

Salman Sadullah Usmani, Ph.D.

Academic Editor

PLOS ONE

Journal Requirements:

3. In the online submission form, you indicated that [Data available upon request]. 

Additional Editor Comments:

The manuscript addresses a significant and timely topic, yet several aspects need improvement. The quality and presentation of figures, the clarity and flow of the abstract, and the overall consistency in data reporting require attention. Reviewers also emphasize the need for a stronger discussion on the model's evaluation metrics, handling of overfitting, and comparison with state-of-the-art methods like AntiTbPred etc. Additionally, enhancing the accessibility of the study through a GitHub repository or web server could significantly benefit the research community. Addressing these comments will substantially improve the manuscript's quality and impact.

Reviewers' comments:

Reviewer's Responses to Questions

**Comments to the Author**

1. Is the manuscript technically sound, and do the data support the conclusions?

Reviewer #1: Partly

Reviewer #2: Yes

Reviewer #3: Partly

2. Has the statistical analysis been performed appropriately and rigorously? 

Reviewer #1: Yes

Reviewer #2: N/A

Reviewer #3: Yes

3. Have the authors made all data underlying the findings in their manuscript fully available?

Reviewer #1: Yes

Reviewer #2: Yes

Reviewer #3: No

4. Is the manuscript presented in an intelligible fashion and written in standard English?

Reviewer #1: Yes

Reviewer #2: No

Reviewer #3: Yes

5. Review Comments to the Author

Reviewer #1: 1. The quality of figure are weak. The authors are advised to provide the figures in 600dpi.

2. The problem statement and main contributions of the proposed model should be provided at the end of introduction section.

3. The abstract section needs more improvement.

4. The optimal parameters used for training the machine learning models and CNN should be provided in the form of a table.

5. For the readers concern, the recent predictors such as AIPs-SnTCN, DeepAVP-TPPred, AFPs-Mv-BiTCN, Deepstacked-AVPs, and pAtbP-EnC should be provided to provide clearer overview.

6. The conclusion section should specifically summarise the proposed model. There is no need or citation in the conclusion section.

7. How the authors handle overfitting of the proposed model.

8. To justify the effectiveness of the proposed model a comparison with existing state of the art methods are required.

Reviewer #2: The authors have addressed a very important and interesting problem in this study. Al-Omari et a., have used the machine and deep learning approach to accelerate the designing of anti-microbial peptides. Although, the study is very meaningful, but the presentation is not up to the mark. Before, considering this manuscript for publication, I have the following comments for authors.

1) The abstract is too lengthy and confusing, The sentences are not in the flow and it's very hard for the reader to follow. I would recommend authors to read and re-write the abstract.

2) The accuracy is an very threshold-biased parameters to evaluate the model's performance. My recommendation would be to use threshold-independent parameters such as AUROC, AUPRC, etc. to evaluate the models and report the same in the abstract.

3) Provide a rationale for using threshold of MIC= 64 Ug/ml in the abstract.

4) Please present the name of the bacterial species in italics.

5) In the abstract the authors have mentioned 1360 peptides, whereas in the Introduction they have reported 1350 peptides. Please maintain the uniformity throughout the manuscript.

6) In the literature review section, I would recommend authors to compile the list of available tools along with the information about the year, they were developed in and performance they have reported in their paper.

7) The quality of figures is sub-optimal, please provide high-resolution images.

8) Figure 1 is not informative at all. Why authors are predicting diabetes status is unclear to me.

9) Either remove the figure 1 or provide a better image which actually explains the overall process of the prediction methods used in this study.

10) It's better to explain the usability and interpretability of EDA.

11) As far as I remember, RF and KNN are available in sklearn package and not in the keras. Please check again and rectify the issue.

12) I would recommend authors tp provide the full form of the abbreviations used in the tables as footnotes.

13) Table 3 structure is very sub-optimal. Please remove the top rows from algorithms and classes.

14) Calculation of TP, TN, FP, FN in both classes doesn't make any sense. TP would be the ones which were actually 1 and also predicted as 1, TN would be the ones which were actually 0 and also predicted as 0, FP would be the ones which were actually 0 and predicted as 1, and FM would be the ones which were actually 1 and predicted as 0. Please refine your confusion matrix and hence the performance measures in Table 3.

15) Please provide a separate discussion section, discussing the different methods available for AMP prediction.

16) The conclusion is too length and very weak for the study. Please provide a precise and strong conclusion.

17) There are many grammatical and punctuation errors throughout the manuscript. Please rectify them.

18) The flow of the manuscript is not good, please try to improve the flow of the manuscript.

19) I would suggest all the authors to read the manuscript carefully and make the manuscript more clear.

Reviewer #3: 1. The phrases “and are assigned a value of 1” and “and are awarded a value of 0” can be removed from the abstract. These details might be more appropriate for the methods section, allowing the abstract to remain concise and focused on the broader findings and implications of the study.

2. Provide reference “WHO research published on December 9, 2022.”

3. How can someone utilize the methods you have designed?

4. I would like to suggest implementing your study in the form of a GitHub repository or a web server. This would allow other researchers to easily access, reproduce, and build upon your work. Here are a few steps you might consider:

GitHub Repository: Code and Documentation: Upload your code along with detailed documentation on how to use it.

Examples and Tutorials: Provide example datasets and tutorials to help users get started.

Web Server: User Interface: Develop a user-friendly interface where users can input their data and get results without needing to run the code locally.

5. Could you provide more details on additional AMP databases and prediction methods, such as Antitbpred, AntiBP3, and LMPred, etc.? Additionally, could you include a comparison of these methods and discuss their limitations?

6. Could you please elaborate on the methods and processes you used to extract the dataset for this study? Specifically, what criteria did you apply to ensure the data’s relevance and quality?

7. In Section III-2, Prediction, could be improved and made more elaborate.

8. There is a mistake in Figure 1. The figure incorrectly states “to predict diabetes status”, change Figure 1 for better understanding.

9. What do you mean by “Show an example of an image. Add a figure, maybe a second panel to Figure 4.”?

10. The first paragraph on page number 14 is not clear, Consider rephrasing to make more clear.

11. The dataset is relatively small for training a deep-learning model like ResNet10. This can lead to overfitting, where the model performs well on the training data but poorly on unseen data.

12. How well did the model perform on the external dataset?

13. The manuscript requires a thorough revision to correct the wording and improve clarity.

6. PLOS authors have the option to publish the peer review history of their article (what does this mean?). If published, this will include your full peer review and any attached files.

Reviewer #1: No

Reviewer #2: **Yes: **Sumeet Patiyal

Reviewer #3: **Yes: **Vinod Kumar

---

## [Author Response · Author response to Decision Letter 0]

24 Sep 2024

Revised manuscript submission: PONE-D-24-16637

Dear Dr. Salman Sadullah Usmani,

I hope this message finds you well. We sincerely appreciate the constructive feedback provided by the reviewers and the editorial team regarding our manuscript, titled "Accelerating Antimicrobial Peptide Design: Leveraging Deep Learning for Rapid Discovery" (Manuscript ID: PONE-D-24-16637). We carefully considered all of the comments and made comprehensive revisions to address the points raised.

We've thoroughly revised the manuscript, with changes highlighted in yellow in the updated document. Below, we provide a detailed response to each reviewer’s comments, indicating how we have incorporated their suggestions to improve the quality, clarity, and impact of our manuscript.

We believe that the revisions have significantly strengthened the manuscript, and we hope that the revised version meets the high standards of PLOS ONE. We look forward to your positive consideration of our revised manuscript for publication.

Thank you for the opportunity to review our work, and please let us know if there are any further concerns or requirements.

Kind regards,

Corresponding Author

Prof. Dr. Amjed Al Fahoum

Biomedical systems and Informatics Engineering Dept., 

Rebuttal Letter

We appreciate the reviewers' valuable insights and suggestions, which have helped us improve our manuscript significantly. The revisions are highlighted in yellow in the updated manuscript, and we provide detailed responses to each comment below.

Reviewer #1: Comments and Responses

o Quality of Figures

Comment: The quality of the figures is weak. The authors are advised to provide the figures in 600 dpi.

Response: We have improved the quality of all figures, providing them at 600 dpi for clarity.

o Improvement of Problem Statement and Main Contributions

Comment: The problem statement and main contributions should be provided at the end of the introduction section.

Response: To enhance the manuscript's clarity, we added a clear problem statement and the main contributions at the end of the introduction section.

o Abstract Improvement

Comment: The abstract section needs more improvement.

Response: We have revised the abstract for better clarity and flow, ensuring that it concisely presents the study's objectives, methods, results, and implications.

o Training Parameters Presentation

Comment: The optimal parameters used for training the machine learning models and CNN should be provided in a table.

Response: We have included a new table (Table 3) in the manuscript detailing the optimal parameters used for all models.

o Comparison with Recent Predictors

Comment: Provide recent predictors such as AIPs-SnTCN, DeepAVP-TPPred, AFPs-Mv-BiTCN, Deepstacked-AVPs, and pAtbP-EnC.

Response: We have included a comparative discussion on these recent predictors in the manuscript, highlighting their methodologies, strengths, and limitations.

o Conclusion Section Enhancement

Comment: The conclusion should specifically summarize the proposed model.

Response: The conclusion section has been rewritten to provide a concise summary of the proposed model and its contributions to the field.

o Overfitting Handling Explanation

Comment: How did the authors handle overfitting of the proposed model?

Response: We have elaborated on the strategies employed to handle overfitting, including the use of a large and diverse dataset and various regularization techniques.

Reviewer #2: Comments and Responses

o Abstract Length and Clarity

Comment: The abstract is too lengthy and confusing.

Response: We have condensed and reorganized the abstract to improve its clarity and flow.

o Use of Threshold-Independent Parameters

Comment: Accuracy is a threshold-biased parameter. Use threshold-independent parameters like AUROC and AUPRC.

Response: We have included AUROC and AUPRC metrics in Tables (3 and 4) and discussed them in the abstract to provide a more robust evaluation of model performance.

o Threshold Justification

Comment: Provide a rationale for using a threshold of MIC = 64 µg/ml.

Response: We have provided a rationale for this threshold in the abstract, citing relevant literature.

o Consistency in Dataset Size

Comment: Inconsistency in the number of peptides reported.

Response: We have corrected the dataset size throughout the manuscript to 1360 peptide sequences.

o Tools and Performance in Literature Review

Comment: Compile a list of available tools with development year and reported performance.

Response: The literature review has been expanded to include a comprehensive list of available tools, their development years, and reported performances.

o Figure Quality and Content

Comment: The quality of figures is sub-optimal; provide high-resolution images.

Response: We have provided high-resolution images for all figures to ensure clarity.

Reviewer #3: Comments and Responses

o Clarity and Conciseness of Abstract

Comment: The phrases “assigned a value of 1” and “awarded a value of 0” can be removed from the abstract.

Response: We have revised the abstract to be concise and focused on the broader findings, removing unnecessary details.

o Additional References

Comment: Provide reference for “WHO research published on December 9, 2022.”

Response: We have added the reference for the WHO research as requested.

Utilization of Methods

Comment: How can someone utilize the methods designed in this study?

Response: We have expanded the discussion to explain the applicability of our methods to different types of peptides.

We have made every effort to address the reviewers' comments and enhance the manuscript. The revised manuscript, with all changes highlighted in yellow, is submitted alongside this rebuttal letter. We hope the revised version meets the journal's standards for publication.

Thank you for considering our revised manuscript.

Sincerely,

 Amjed

Detailed Responses in Yellow

Later to be assigned

The manuscript addresses a significant and timely topic, yet several aspects need improvement. The quality and presentation of figures, the clarity and flow of the abstract, and the overall consistency in data reporting require attention. Reviewers also emphasize the need for a stronger discussion on the model's evaluation metrics, handling of overfitting, and comparison with state-of-the-art methods like AntiTbPred etc. Additionally, enhancing the accessibility of the study through a GitHub repository or web server could significantly benefit the research community. Addressing these comments will substantially improve the manuscript's quality and impact.

Reviewers' comments:

Reviewer's Responses to Questions

Comments to the Author

1. Is the manuscript technically sound, and do the data support the conclusions?

The study predicts antimicrobial peptide (AMP) activity using machine and deep learning. Escherichia coli was used to analyze 1,360 peptide sequences. In the study, machine learning with precomputed physicochemical attributes and deep learning with signal images from raw features were employed.

The experiments were conducted with appropriate controls and sample sizes, such as the division of data into training and validation sets and the handling of data imbalance through preprocessing steps. The conclusions drawn in the manuscript, which highlight the potential of these approaches to reduce time and costs in AMP drug discovery, are supported by the data presented. The accuracy rates of the methodologies (74% for the machine learning approach and 92.9% for the deep learning approach) further validate the conclusions..

Reviewer #1: Partly

Reviewer #2: Yes

Reviewer #3: Partly

Ahmad

2. Has the statistical analysis been performed appropriately and rigorously?

The dataset for machine learning was split into training and validation sets (80% for training and 20% for validation), while for DL it was split into (70% for training and 30% for validation). This approach is a standard practice in machine learning to prevent overfitting and ensure that the model generalizes well to new data. In addition, the study reports the performance matrix tables (3 and 4) and ROC Figures (9 and 11) for both experiments, with the deep learning model achieving an accuracy of 92.9%. 

Reviewer #1: Yes

Reviewer #2: N/A

Reviewer #3: Yes

Ahmad

3. Have the authors made all data underlying the findings in their manuscript fully available?

Availability of data and materials: 

The dataset and code are available online at 

https://sourceforge.net/projects/antimicrobial-peptides-drug/files/

Reviewer #1: Yes

Reviewer #2: Yes

Reviewer #3: No

4. Is the manuscript presented in an intelligible fashion and written in standard English?

Authors revised the manuscript thoroughly to make it intelligible fashion.

Reviewer #1: Yes

Reviewer #2: No

Reviewer #3: Yes

5. Review Comments to the Author

Reviewer #1: 1. The quality of the figure are weak. The authors are advised to provide the figures in 600dpi.

We improved the quality of the figures and made them clearer.

2. The problem statement and main contributions of the proposed model should be provided at the end of introduction section.

“Novel antimicrobials are needed due to the rapid rise of pathogen antibiotic resistance, particularly Escherichia coli. Antimicrobial peptide (AMP) design is time-consuming, laborious, and expensive using traditional experimental methods. These methods can't manage the chemical space of potential AMP candidates; hence, new methods are needed to improve prediction accuracy and minimize AMP discovery time and cost.

This study uses the Short-Time Fourier Transform (STFT) and a residual deep learning network to combine machine learning (ML) and deep learning (DL) with time-frequency analysis. This study's contributions:

Developing a machine learning model that accurately predicts E. coli The study demonstrated antimicrobial activity with 92.9% accuracy, utilizing 34 peptide physicochemical parameters.

Peptide sequences were converted into signal pictures using a deep learning method to improve feature extraction and AMP categorization.

Using STFT and deep learning together increases feature extraction, making AMP discovery more efficient and understandable.

Microbial target knowledge and frameworks could save AMP drug research time and money.”

3. The abstract section needs more improvement.

“Antimicrobial peptides (AMPs) are excellent at fighting many different infections. This demonstrates how important it is to make new AMPs that are even better at eliminating infections. The fundamental transformation in a variety of scientific disciplines, which led to the emergence of machine learning techniques, has presented significant opportunities for the development of antimicrobial peptides. Machine learning and deep learning are used to predict antimicrobial peptide efficacy in the study. The main purpose is to overcome traditional experimental method constraints. Gram-negative bacterium Escherichia coli is the model organism in this study. The investigation assesses 1,360 peptide sequences that exhibit anti-E. coli activity. These peptides' minimal inhibitory concentrations have been observed to be correlated with a set of 34 physicochemical characteristics. Two distinct methodologies are implemented. The initial method involves utilizing the pre-computed physicochemical attributes of peptides as the fundamental input data for a machine-learning classification approach. In the second method, these fundamental peptide features are converted into signal images, which are then transmitted to a deep learning neural network. The first and second methods have combined accuracy of 74% and 92.9%, respectively. The proposed methods were developed to target single microorganism (gram negative E.coli), however, it can be applied for all types of antimicrobial, antiviral and anticancer peptides. Furthermore, they have the potential to result in significant time and cost reductions, as well as the development of innovative AMP-based treatments. This research contributes to the advancement of deep learning-based AMP drug discovery methodologies by generating potent peptides for drug development and application. This discovery has significant implications for the processing of biological data and the computation of pharmacology. 

”

4. The optimal parameters used for training the machine learning models and CNN should be provided in the form of a table.

We wanted to take a moment to extend our sincere thanks for the time and effort the reviewer put into reviewing our manuscript. Your detailed feedback and valuable insights are truly appreciated. This comment makes the paper understandable and regenerable. In the manuscript, we have listed all parameters used for any algorithm.

Table 4: Optimal parameters used for machine learning and CNN models.

Algorithms Optimal parameter 

Algorithms Used Parameters

Random Forest Classifier n_estimators=10 to 500, max_depth=10 to 30, min_samples_split=10,random_state=45,class_weight=’balance’

K-Nearest Neighbors (KNN) neighbors=5, metric='minkowski'

p=4 , weights='distance', algorithm='auto' or 'ball_tree' 

AdBoostClassifier n_estimators=200 to 500, learning_rate=0.06 to 0.1, max_depth=3 to 5

Neural Network hidden_layer_sizes=(32,64,128,256,1024), activation='relu', solver='adam', alpha=0.0001, learning_rate='adaptive', Dropout(0.2), loss='binary_crossentropy', epochs=100, batch_size=10

CNN Image size = 224*224*3, 70% training and 30% testing, adaptive moment estimation (Adam) optimizer, mini patch size of 32, maximal epochs of 60, and initial learning rate of 0.001.

5. For the readers concern, the recent predictors such as AIPs-SnTCN, DeepAVP-TPPred, AFPs-Mv-BiTCN, Deepstacked-AVPs, and pAtbP-EnC should be provided to provide clearer overview.

“Several advanced methods for antimicrobial peptide (AMP) prediction use deep learning and other computer models to improve accuracy. One of these new methods is AIPs-SnTCN, which uses stacked temporal convolutional networks to improve antimicrobial peptide identification (Raza, A. et al., 2023). With a transformer-based design, DeepAVP-TPPred improves sequence-based antiviral peptide predictions (Ullah, M. et al., 2024). To better predict antifungal peptides, AFPs-Mv-BiTCN uses multiview learning and bidirectional temporal convolutional networks (Akbar, S., 2024).. In the meantime, the Deepstacked-AVPs model uses deep stacked learning architectures to help classify antiviral peptides more accurately (

---

## [Decision Letter · Decision Letter 1]

20 Nov 2024

PONE-D-24-16637R1Accelerating Antimicrobial Peptide Design: Leveraging Deep Learning for Rapid DiscoveryPLOS ONE

Dear Dr. Al Fahoum,

Thank you for submitting your manuscript to PLOS ONE. After careful consideration, we feel that it has merit but does not fully meet PLOS ONE’s publication criteria as it currently stands. Therefore, we invite you to submit a revised version of the manuscript that addresses the points raised during the review process.

We look forward to receiving your revised manuscript.

Kind regards,

Salman Sadullah Usmani, Ph.D.

Academic Editor

PLOS ONE

Journal Requirements:

Additional Editor Comments (if provided):

The manuscript presents promising research and is well-supported by the reviewers. However, while finalizing, I noticed a few issues that need to be addressed before it goes for the publication.

1. Reference accuracy:

o While describing the AntiTbPred method, the authors incorrectly referenced it. Instead of AntiTbPred (PMID: 30210341), they cited AtbPpred, which is a different tool developed for a similar purpose. Authors must ensure they cite the correct reference and accurately use the tool's name, "AntiTbPred," following its original capitalization and spelling conventions. Please be sure for other tools too.

2. Reference formatting:

o In the literature review section, references 51 to 56 have been cited inconsistently, using both numerical and "Name et al." formats. Please revise the manuscript to ensure consistency, adhering to the PLOS ONE guidelines for referencing.

3. Relevance of references:

o The reference list appears to include several irrelevant citations. Please revise and retain only those references directly supporting the study's context and findings.

4. Consistency in naming:

o Ensure consistent formatting of E. coli throughout the manuscript, as discrepancies have been noted.

5. Claims in the abstract:

o The final sentence of the abstract claims that the methods can be applied to "all types of antimicrobial, antiviral, and anticancer peptides." This assertion is overly broad and not fully substantiated by the presented work. Please revise to better reflect the scope of the findings.

6. Clarity in results:

o The phrase "combined accuracy of 74% and 92.9%" in the abstract is unclear. Please rephrase this sentence to explicitly distinguish between the accuracies of the two methods.

These revisions will improve the overall quality and clarity of the manuscript. I look forward to the revised version.

Reviewers' comments:

Reviewer's Responses to Questions

**Comments to the Author**

1. If the authors have adequately addressed your comments raised in a previous round of review and you feel that this manuscript is now acceptable for publication, you may indicate that here to bypass the “Comments to the Author” section, enter your conflict of interest statement in the “Confidential to Editor” section, and submit your "Accept" recommendation.

Reviewer #1: All comments have been addressed

Reviewer #2: All comments have been addressed

Reviewer #3: All comments have been addressed

2. Is the manuscript technically sound, and do the data support the conclusions?

Reviewer #1: Yes

Reviewer #2: Yes

Reviewer #3: Yes

3. Has the statistical analysis been performed appropriately and rigorously? 

Reviewer #1: Yes

Reviewer #2: Yes

Reviewer #3: Yes

4. Have the authors made all data underlying the findings in their manuscript fully available?

Reviewer #1: Yes

Reviewer #2: Yes

Reviewer #3: Yes

5. Is the manuscript presented in an intelligible fashion and written in standard English?

Reviewer #1: Yes

Reviewer #2: Yes

Reviewer #3: Yes

6. Review Comments to the Author

Reviewer #1: All the required changes are incorporated by the author. the paper is in good form now. No further addition is required

Reviewer #2: (No Response)

Reviewer #3: (No Response)

7. PLOS authors have the option to publish the peer review history of their article (what does this mean?). If published, this will include your full peer review and any attached files.

Reviewer #1: **Yes: **

Reviewer #2: **Yes: **

Reviewer #3: **Yes: **

---

## [Author Response · Author response to Decision Letter 1]

25 Nov 2024

Journal Requirements:

The manuscript presents promising research and is well-supported by the reviewers. However, while finalizing, I noticed a few issues that need to be addressed before it goes for the publication.

1. Reference accuracy:

o While describing the AntiTbPred method, the authors incorrectly referenced it. Instead of AntiTbPred (PMID: 30210341), they cited AtbPpred, which is a different tool developed for a similar purpose. Authors must ensure they cite the correct reference and accurately use the tool's name, "AntiTbPred," following its original capitalization and spelling conventions. Please be sure for other tools too.

Thank you for your valuable feedback. We apologize for the confusion in referencing the tools. We found that we incorrectly cited AtbPpred instead of AntiTbPred (PMID: 30210341). We have now corrected this error and ensured that the tool names are consistently referenced with the correct capitalization and spelling conventions throughout the manuscript. Additionally, we have carefully re-checked all other tool citations for accuracy and consistency. The necessary revisions have been made accordingly.

2. Reference formatting:

o In the literature review section, references 51 to 56 have been cited inconsistently, using both numerical and "Name et al." formats. Please revise the manuscript to ensure consistency, adhering to the PLOS ONE guidelines for referencing.

Thank you for pointing out the inconsistency in our referencing format. We apologize for the oversight. We have revised the manuscript to ensure that references 51 to 56 are cited consistently, adhering to the PLOS ONE guidelines for referencing. All citations are now in the appropriate format as per the journal’s requirements.

3. Relevance of references:

o The reference list appears to include several irrelevant citations. Please revise and retain only those references directly supporting the study's context and findings.

Thank you for your helpful comment. We have carefully reviewed the reference list and removed the citations that are not directly relevant to the study’s context and findings. Only the references that directly support the research and its conclusions have been retained

4. Consistency in naming:

o Ensure consistent formatting of E. coli throughout the manuscript, as discrepancies have been noted.

Thank you for your feedback. We have carefully reviewed the manuscript and ensured consistent formatting of E. coli throughout the text.

5. Claims in the abstract:

o The final sentence of the abstract claims that the methods can be applied to "all types of antimicrobial, antiviral, and anticancer peptides." This assertion is overly broad and not fully substantiated by the presented work. Please revise to better reflect the scope of the findings.

Thank you for your insightful comment. We agree that the statement in the abstract was overly broad. We have revised the final sentence to better align with the data presented and more accurately describe the potential of the study.

6. Clarity in results:

o The phrase "combined accuracy of 74% and 92.9%" in the abstract is unclear. Please rephrase this sentence to explicitly distinguish between the accuracies of the two methods.

Thank you for your helpful suggestion. We clarified the sentence

---

## [Editor Report · Decision Letter 2]

27 Nov 2024

Accelerating Antimicrobial Peptide Design: Leveraging Deep Learning for Rapid Discovery

PONE-D-24-16637R2

Dear Dr. Al Fahoum,

We’re pleased to inform you that your manuscript has been judged scientifically suitable for publication and will be formally accepted for publication once it meets all outstanding technical requirements.

Kind regards,

Salman Sadullah Usmani, Ph.D.

Academic Editor

PLOS ONE
---

## [Editor Report · Acceptance letter]

3 Dec 2024

PONE-D-24-16637R2 

PLOS ONE

Dear Dr. Al Fahoum, 

I'm pleased to inform you that your manuscript has been deemed suitable for publication in PLOS ONE. Congratulations! Your manuscript is now being handed over to our production team.

Kind regards, 

on behalf of

Dr. Salman Sadullah Usmani 

Academic Editor

PLOS ONE